# *Euphorbia royleana* Boiss Derived Silver Nanoparticles and Their Applications as a Nanotherapeutic Agent to Control Microbial and Oxidative Stress-Originated Diseases

**DOI:** 10.3390/ph16101413

**Published:** 2023-10-04

**Authors:** Rehman Ullah, Saiqa Afriq Jan, Muhammad Nauman Khan, Moona Nazish, Asif Kamal, Alevcan Kaplan, Hany M. Yehia, Khaloud Mohammed Alarjani, Rashad Alkasir, Wajid Zaman

**Affiliations:** 1Pharmacognosy Laboratory, Department of Botany, Faculty of Life and Environmental Sciences, University of Peshawar, Peshawar 25120, Pakistan; saiq.12@gmail.com; 2Department of Botany, Islamia College Peshawar, Peshawar 25120, Pakistan; 3University Public School, University of Peshawar, Peshawar 25120, Pakistan; 4Department of Botany, Rawalpindi Women University, Rawalpindi 46300, Pakistan; mnbot@f.rwu.edu.pk; 5Department of Plant Sciences, Faculty of Biological Sciences, Quaid-i-Azam University, Islamabad 45320, Pakistan; kamal@bs.qau.edu.pk; 6Department of Crop and Animal Production, Sason Vocational School, Batman University, Batman 72060, Turkey; kaplanalevcan@gmail.com; 7Food Science and Nutrition Department, College of Food and Agricultural Sciences, King Saud University, P.O. Box 2451, Riyadh 11451, Saudi Arabia; hanyehia@ksu.edu.pk; 8Department of Botany and Microbiology, College of Science, King Saud University, Riyadh 11451, Saudi Arabia; kalarjani@ksu.edu.sa; 9CAS Key Laboratory of Pathogenic Microbiology and Immunology, Institute of Microbiology, Chinese Academy of Sciences, Beijing 100101, China; 10Department of Life Sciences, Yeungnam University, Gyeongsan 38541, Republic of Korea

**Keywords:** *Euphorbia royleana*, AgNPs, SEM, antioxidant, hepatoprotective

## Abstract

Nanotechnology is one of the most advance and multidisciplinary fields. Recent advances in nanoscience and nanotechnology radically changed the way we diagnose, treat, and prevent various diseases in all aspects of human life. The use of plants and their extracts is one of the most valuable methods towards rapid and single-step protocol preparation for various nanoparticles, keeping intact “the green principles” over the conventional ones and proving their dominance for medicinal importance. A facile and eco-friendly technique for synthesizing silver nanoparticles has been developed by using the latex of *Euphorbia royleana* as a bio-reductant for reducing Ag+ ions in an aqueous solution. Various characterization techniques were employed to validate the morphology, structure, and size of nanoparticles via UV–Vis spectroscopy, XRD, SEM, and EDS. FTIR spectroscopy validates different functional groups associated with biomolecules stabilizing/capping the silver nanoparticles, while SEM and XRD revealed spherical nanocrystals with FCC geometry. The results revealed that latex extract-mediated silver nanoparticles (LER-AgNPs) exhibited promising antibacterial activity against both gram-positive and -negative bacterial strains (*Bacillus pumilus*, *Staphylococcus aureus*, *E. coli*, *Pseudomonas aeruginosa*, and *Streptococcus viridians*). Both latex of *E. royleana* and LER-AgNPs were found to be potent in scavenging DPPH free radicals with respective EC50s and EC70s as 0.267% and 0.518% and 0.287% and 0.686%. ROSs produced in the body damage tissue and cause inflammation in oxidative stress-originated diseases. H_2_O_2_ and OH* scavenging activity increased with increasing concentrations (20–100 μg/mL) of LER-AgNPs. Significant reestablishment of ALT, AST, ALP, and bilirubin serum levels was observed in mice intoxicated with acetaminophen (PCM), revealing promising hepatoprotective efficacy of LER-AgNPs in a dose-dependent manner.

## 1. Introduction

Nano-biotechnology is a new area in medicine that makes use of nano-sized materials for targeted cell-related, or specifically for tissue-related, medicinal interventions. Nanotechnology aims to develop and apply methodologies to manufacture nano-systems that can interact with high specificity at the molecular level in order to obtain maximal therapeutic effectiveness with minimum negative effects [1,2]. Nanotechnology has emerged as the main tool in the bio-production process of metallic nanoparticles (MNPs). In past decades, nanotechnology has undergone several advances. In comparison to other sectors, the significance of agricultural research technology is relatively new. The use of nanoparticles as nanofertilizers is one of its important roles in useful and effective crop production. When employed at the appropriate concentration, these nanofertilizers significantly boost crop yield, plant growth, and plant tolerance [3]. Nowadays, the work is in process to discover new applications with great technological potential. Smaller size with greater potential is the saying, as far as the Nanoworld is concerned. This has led to the generation of nanoparticles of different sizes and shapes.

Different chemical methodologies have been utilized for nanoparticle synthesis in the field of nanotechnology [4]; however, it is true that biological catalysts, such as microorganisms and plants, have been used to synthesize stable particles of smaller sizes in less time [5]. Nanoparticles have distinct physicochemical features that are used for a variety of applications in biological sciences, chemical engineering, medicine, and agriculture [6]. Metal ions are decreased via biological systems, resulting in nanoparticles with more potential than their larger counterparts. Gold, silver, platinum, zinc, and other metals have been used in the biosynthesis of nanoparticles [7,8], with plant extracts being utilized in some cases. The extracts are vital in both decreasing the ions to nano size and capping the nanoparticles [9]. The use of plant extract for nanoparticle production is preferable to other biological procedures because it eliminates the time-consuming process of maintaining cell cultures and can be scaled up for large-scale preparations.

Phytochemicals such as phenolic compounds, terpenoids, and alkaloids have been discovered to be effective reducing agents [1,10]. All of these have been reported to have a faster synthesis rate when compared to other nanoparticle formation methods. Plants, on the other hand, are known to contain a variety of phytochemicals that may be responsible for the formation of these nanostructures, and the chemical interactions that result in the formation of these nanoparticles are currently being researched [11]. Many medicinal plants have antibacterial qualities; hence, testing for antibacterial activity of plant extracts used to make nanoparticles might be performed to investigate the combined effect of the metal and the plant extract. *E. royleana* is a medicinal plant from the *Euphorbiaceae* family, which comprises both wild and cultivated species of herbs, shrubs, trees, and succulent plants that are primarily found in tropical and temperate zones [1]. It can be found between 3000 and 5000 m elevation in subtropical rain shadow valleys and dry slopes of the Himalayan range in India, Nepal, Bhutan, Pakistan, Indonesia, Taiwan, Myanmar, Yunnan, and China [12]. *E. royleana* plays a role in its ecosystem by influencing plant communities, providing habitat and food sources for various organisms, and contributing to soil stabilization. However, its allelopathic properties, potential invasiveness, and toxicity can also have ecological implications that need to be carefully considered in its natural habitat. Numerous researchers have underscored the significance of *E. royleana* in its interactions with various organisms, such as insects andviruses, including its effects against the ridge gourd mosaic virus, fungi, and nematodes [13]. The *Euphorbiaceae* family produces milky latex that has traditionally been used to treat skin issues, asthma, jaundice, anemia, cough, and constipation [13,14]. They also include a wide spectrum of secondary metabolites, including tannins, glycosides, alkaloids, steroids, flavonoids, and terpenoids [15]. In experimental animals, *E. royleana* latex has strong anti-inflammatory and anti-arthritic efficacy [16]. Despite the fact that *E. royleana* is rich in secondary metabolites and has a number of conventional and therapeutic uses, it has not been investigated for the production of NPs [17]. According to several studies, combining metal ions with plant extracts improves the biological potential of nanoparticles or extracts due to the synergistic effect of both. The observed biocompatibility of green-synthesized AgNPs hints at their potential as therapeutic agents in the realm of healthcare. Given that AgNPs are already employed in cosmetics and various health-related applications, the utilization of biogenic AgNPs may instill greater confidence in the safety and efficacy of these products. Therefore, the present study is focused on the synthesis of silver nanoparticles using *E. royleana* latex as a reducing and surface-functionalizing agent, as well as characterizations and antibacterial, hepatoprotective, and antioxidant efficacy evaluation.

## 2. Results

### 2.1. Nanoparticle Synthesis and Characterization

Phytogenic AgNPs were synthesized by mixing 0.1 mM aqueous solution of silver nitrate with LER, with continuous stirring for 5–10 min at 40 ± 3 °C. The milky white color of the reaction mixture turned dark brown due to the excitation of surface plasmon resonance (SPR) vibrations of the AgNPs synthesized in the reaction medium. UV-Vis spectral analysis of LER-AgNPs showed a characteristic absorption peak around 440 nm due to oscillation of the surface plasmon of silver nanoparticles in resonance with incident photons (Figure 1). Using Mie theory, the size of the LER-AgNPs was calculated as 76 nm using UV-Vis spectrographs.

X-ray diffraction analysis showed diffraction intense peaks at 2θ = 38.09, 48.32, and 76.64, which can be indexed to (111), (200), and (300) planes of the crystalline phase with fcc dimensions of the AgNPs (Figure 2). The Debye–Scherrer equation was used to calculate the crystalline size based on a FWHM of 38 nm.

The SEM images presented spherical morphology with an average particle size ranging from 8–200 nm (Figure 3). The newly synthesized AgNPs with large surface energy conjoined, due to the van der Waals forces form agglomeration, to create thermodynamically relatively stable bulk particles.

FTIR spectrum of LER showed several absorption peaks at 1022, 2864, 2346, 1845, 1649, and 3227 cm^−1^ associated with C–0 stretch (carbohydrates, glucose, and fructose), C-H (Lipid), C≡C stretch Alkynes (propyne), C≡C stretching (lipid and fatty acids), C=O stretch Amides (methane amid), and hydrogen-bonded O-H stretch asymmetrical, respectively (Figure 4). The reduction in bands at 1022, 2864, 2346, 1845, 1649, and 3227 cm^−1^ in the AgNPs may be attributed to the reduction of Ag^+^ to Ag^0^ as the core of nanoparticles (Figure 3). We suggested that the synthesized AgNPs were capped by biomolecules with the functional groups sitolar as Skimmidol, which was responsible for the fabrication of the silver ions (Figure 1).

EDS characterization was carried out for elemental mapping of LER-AgNPs and shows a strong metallic silver signal around 2.3 Kev. The Ag ED spectrograph showing signals for calcium, chlorine, sodium, magnesium, silicon, sulfur, etc. is due to the X-ray emission from the biomolecules of LER-capping AgNPs (Figure 5).

Dynamic light scattering (DLS) spectrogram analysis was employed to characterize the LER-AgNPs. The DLS spectrogram (Figure 6a) revealed a uniform particle size and unimodel size distribution with an average hydrodynamic nanoparticle diameter of 85.36 nm. The Zeta potential measurements (Figure 6b) indicated a highly negative surface charge of −25.5 mV, underscoring the excellent stability of the LER-AgNPs in dispersion. These findings collectively suggest that the synthesized silver nanoparticles exhibit not only a consistent size profile but also possess a surface charge conducive to various potential applications.

### 2.2. DPPH Scavenging Activity

The reducing effect of LER and LER-AgNPs on DPPH was analyzed spectroscopically. A dose- and time-dependent DPPH inhibition by LER and LER-AgNPs is presented in Figure 7A,B. The antioxidant potential falls in the order of ascorbic acid (A.A) > LER > AgNPs, whereas the dose-wise pattern recorded was 1 mg/mL < 2 mg/mL < 5 mg/mL. The duration pattern falls as 90 min > 60 min > 30 min. EC50s and EC70s of DPPH scavenging by LER (0.267% and 0.518%), AgNPs (0.287% and 0.686%), and ascorbic acid (0.039% and 0.082%) at 30 min of incubation were recorded (Figure 7B).

### 2.3. Hydroxyl Scavenging Potential

The antioxidant activity of LER and LER-AgNPs was evaluated using an OH free radical scavenging assay. This assay shows the abilities of the latex and ascorbic acid to inhibit hydroxyl radical-mediated deoxyribose degradation in an Fe^3+^–EDTA–ascorbic acid and H_2_O_2_ reaction mixture. The Student’s *t*-test showed the significant (*p* < 0.05) and dose-dependent *OH scavenging efficacy of both LER and AgNPs compared to untreated dye. The percent *OH scavenging potential of LER, AgNPs, and ascorbic acid is visualized in Figure 8, showing that LER and AgNPs inhibit phenol red oxidation by scavenging hydroxyl radicals by 43.5%, 48.4%, 49.6%, 58.1%, and 67.1% and 36.65%, 41.3%, 44.4%, 52.3%, and 60.5%, at 20, 40, 60, 80, and 100 μg/mL concentrations, respectively (Figure 8).

### 2.4. Hydrogen Peroxide Scavenging 

The scavenging ability of LER and LER-AgNPs on hydrogen peroxide is shown (Figure 9). The LER and AgNPs were capable of scavenging hydrogen peroxide in a dose-dependent manner. The ascorbic acid exhibited maximum scavenging of hydrogen peroxide by 88.33% at a maximum experimental dose of 100 µg/mL. LER scavenged H_2_O_2_ by 31.06%, 41.97%, 47.71%, 56.21%, and 61.10%, while AgNPs did so by 35.33%, 37.63%, 41.22%, 47.19%, and 57.93%, at respective experimental doses of 20, 40, 60, 80, and 100 μg/mL, hence showing a more pronounced antioxidant effect of LER than AgNPs (Figure 9). Although hydrogen peroxide is not particularly reactive, it can be hazardous to cells when it produces the hydroxyl radical in the cells [18] (Kumar et al., 2020). As a result, eliminating H_2_O_2_ from cell or dietary systems is critical for antioxidant defense.

### 2.5. Antibacterial Activity

It has been known that silver nanoparticles (AgNPs) can inhibit microbial growth and even kill microbes. LER and LER-AgNPs were used against five different bacterial strains (*Bacillus pumilus*, *E. coli*, *Staphylococcus aureus*, *Pseudomonas aeruginosa*, and *Streptococcus viridians*). The effects of LER and AgNPs on bacterial growth have been studied by employing the disc diffusion method, which is quite comparable with the standard antibiotics (streptomycin), as shown in (Figure 10). LER exhibit maximum growth inhibition against *Streptococcus viridians* (67.30% of streptomycin), and the minimum activity was shown against *Bacillus pumilus* (34.16% of streptomycin), while LER-AgNPs showed the maximum inhibitory potential of *Streptococcus viridians* (104.21% of streptomycin), and minimum growth inhibition was reported against *Staphylococcus aureus* (68.99% of streptomycin).

### 2.6. Antifungal Activity

LER and LER-AgNPs were evaluated for their fungicidal potential against *Aspergillus flavus* Link and *Aspergillus parasiticus* Speare. LER at 1000 µg/mL exhibited no antifungal activity against either fungal strain used, while, at 2000 µg/mL, 55.44% of mycelium growth inhibition was reported in *Aspergillus flavus*. LER-AgNPs at 1000 µg/mL revealed evident antifungal activity against both *Aspergillus parasiticus* (28.83%) and *Aspergillus flavus* (48.04%), while, at 2000 µg/mL, the mycelial growth inhibition observed was 28.67% in *Aspergillus flavus* and 73.15% in *Aspergillus parasiticus* (Figure 11).

### 2.7. Hepatoprotective Activity

The hepatoprotective efficacy of LER-AgNPs was determined by analyzing liver functionality biomarkers (ALT, ALP AST, bilirubin, and protein) and histopathological examination. The paracetamol-administered group had enhanced the levels of ALP, ALT, and bilirubin (both total and direct bilirubin levels), while AST and protein levels decreased significantly. On the other hand, treatment with silymarin at 10 mg/kg of body weight (standard) and LER-AgNPs (100, 200, and 200 mg/kg BW, p.o) considerably stabilized the raised levels of ALP, ALT, and bilirubin and also significantly improved the declined AST and protein levels in experimental animals. Briefly, the groups treated with LER-AgNPs (at 100, 200, and 200 mg/kg BW) had significantly reduced AST (112, 96, and 92 IU/L, respectively), ALP (208, 192, and 187, IU/L, respectively), and bilirubin (0.97, 0.94, and 0.86 g/dL) levels but increased protein levels (1.87, 2.08, and 2.81 g/dL) compared to the PCM-administered group (Table 1). Moreover, the histopathological study of the liver of PCM-intoxicated animals revealed necrotic lesions caused by acetaminophen toxicity. The PCM-intoxicated animals treated with silymarin and LER-AgNPs potentially healed heat necrosis (Figure 12).

### 2.8. Gas Chromatography–Mass Spectrometry (GC–MS) Analysis

The gas chromatography–mass spectrometry (GC–MS) analysis was performed to elucidate the composition of the latex of *E. royleana*. The analysis was conducted at the centralized Resource Laboratory (CRL) at the University of Peshawar, Pakistan, using a GC–MS “model GC: 7890B, MS: 5977B Aligent technologies USA”. Retention time (3–35 min) data revealed the presence of 18 different peaks (Figure 13) attributed different compounds in the sample, as shown in Table 2. The minimal RT value (17.437 min) was recorded for n-Hexadecanoic acid, followed by 18.914 min for Methyl 9-cis,11-trans-octadecadienoate. Obtusifoliol was observed at the maximal RT value (31.541 min). Most of the compounds were from fatty acid, their derivatives, and derived lipid class. 6-Octadecenoic acid, Ricinoleic acid, and (9E,11E)-Octadecadienoic acid were observed to be the dominant phytochemicals with the maximal peak area (%) of 36.83%, 15.25% and 12.37%, respectively (Table 2).

## 3. Discussion

During the synthesis of AgNPs, the milky white color of the reaction mixture of LER and silver nitrate changed to brown due to the collective oscillations of electrons restricted to the surface of the newly synthesized AgNPs. It was investigated that silver nanoparticles were mostly spherical in shape and polydisperse in nature [19,20]. Comparative IR spectroscopy of LER and LER-AgNPs confirmed the presence of different functional groups adsorbed on the surface of AgNPs. FTIR analysis confirmed the involvement of functional groups in biomolecules stabilizing the suspension of silver nanoparticles and capping the metal nanoparticles [21]. The absorption of a strong silver signal, as well as other elements attached to the surface of silver nanoparticles, was discovered using EDX [22]. The Zeta potential (ZP) of LER-AgNPs was recorded as −25.5 mV (Figure 6b), where particles with ZP values ranging from ±25 to ±50 mV are regarded as exceptionally stable. However, particles that fall outside of this ZP range tend to exhibit significant instability [23]. The hydrodynamic size of the synthesized LER-AgNPs was also assessed through DLS spectroscopy, as depicted in Figure 6A. The size distribution of these particles displayed a narrow span with an average hydrodynamic diameter of 85.36 nm. It is worth noting that the DLS-measured size of LER-AgNPs was unexpectedly smaller compared to that observed in the SEM images (Figure 3). This difference may arise from the fact that SEM images represent only a tiny fraction of the sample while DLS analyzes a larger number of particles, or it could be due to potential core diameter reduction caused by the formation of a permeable polymer layer in contact with water during the DLS measurements [24].

Phenolic compounds and flavonoids present in plant extract are related to resilient antioxidant potential, and they possess biological activities. For the antioxidant test, the DDPH was used as a free radical and showed good absorption of light at 517 nm. Due to its strong scavenging potential, ascorbic acid is generally preferred as the standard [25]. The LER-AgNPs could have reacted with the nitrogen and oxygen atoms of the free radicles and converted them into less non-toxic or less toxic compounds that could enhance cell viability [26,27]. The LER and synthesized LER-AgNPs demonstrated antioxidant properties against DPPH, hydrogen peroxide, and hydroxyl free radicals. Both LER and LER-AgNPs exhibited significant DPPH radical scavenging activity (EC50 = 266.81, 114.78, and 77.51 and 286.52, 124.80, and 82.84 μg/mL, respectively, after 30, 60, and 90 min of incubation). The scavenging potential might be attributed to the synergic effect of both the elemental silver and phytochemicals present in LER and the dopping surface of AgNPs [28,29]. These findings align with previous research that has reported the DPPH [2,30,31,32] scavenging potential of Euphorbia species and AgNPs coated with plant constituents. The AgNPs’ significant antibacterial activity is owed to their affinity for cell membrane surface proteins, which disrupts cell permeability. AgNPs bind to sulfur-containing proteins in the cell membrane, producing membrane damage and depletion of the microorganism’s intracellular ATP levels [33]. Heavy metals are thought to react with proteins by combining the thiol (SH) groups, which causes the proteins to become inactive [34].

Silver nanoparticles can easily access the nuclear content of bacteria due to their unique size and larger surface area [35]. It has been investigated that AgNPs interfere with fungal cell walls, as well as cell membranes, eventually inhibiting fungal growth. Similarly, AgNPs also form complexes with nitrogenous bases in DNA, inhibiting cellular metabolism and acting as a potent inhibitor of fungal DNA. Hydroxyl radicals are produced in the body via a variety of metabolic events. They are short-lived, highly hazardous free radicals with a preference for biomolecules such as lipids, proteins, amino acids, sugars, and deoxyribonucleic acids, which can cause cancer, mutagenesis, and cytotoxicity [36]. In this work, LER and LER-AgNPs significantly scavenged DPPH, H_2_O_2_, and OH free radicals. Polyphenols, flavonoids, amides, etc. are present in Euphorbia, and latex-capping AgNPs act as electron donors, thus neutralizing the redox potential of ROS. Also reported is the dose-dependent hydrogen peroxide scavenging activity of *Cola nitida* extract and its fabricated AgNPs [37]. Drug-induced hepatotoxicity is a major medical issue. Consequently, these agents cause enzymatic releases, such as ALP, AST, and ALT into the blood, which is allied with the overproduction of reactive oxygen species [38]. Earlier studies suggest that exposure to LPS resulted in significant damage to hepatic tissue, characterized by increased levels of cytokines and proinflammatory markers. Silver nanoparticles demonstrated notable effectiveness in reducing LPS-induced liver injury, primarily by maintaining cytokine levels and suppressing inflammatory markers such as NO, Cox2, TNF-α, and IL-6 [39]. Elevation of blood transaminases serves as an indicator of liver structural impairment. Exposure to PCM affects subcellular structures within the liver, including the cell membrane, mitochondria, endoplasmic reticulum, and Golgi apparatus [40,41]. This toxic impact leads to the release of cytoplasmic enzymes into the bloodstream as a consequence of liver damage. Consequently, both increased cell membrane permeability and elevated enzyme activity contribute to structural liver injury [41,42]. The rise in circulating liver enzyme levels results from the leakage of serum enzymes due to lipid peroxidation. In our investigation, we observed an elevation in the levels of serum enzymes such as ALT, AST, and ALP, which is indicative of liver damage. The reduction of serum transaminases to nearly normal levels following treatment with LER-AgNPs suggests the potential regeneration of hepatocytes and a possible healing effect on the hepatic parenchyma. The overall improvement observed across treatment groups III–VI underscored the therapeutic impact of the silymarin and LER-AgNPs. Specifically, treatment with HD of LER-AgNPs (300 mg/kg BW, Group VI) demonstrated effectiveness in terms of all the evaluated biochemical parameters (Table 2).

## 4. Material and Methods

### 4.1. Preparation of Plant Extract

Extracts of *E. royleana* latex were used for the nanoparticle synthesis. Stem of the *E. royleana* was sutured with sharp blade and the latex was obtained. The collected latex was blended in 500 mL of 70% ethanol (*v*/*v*). These mixtures were placed in a shaking incubator for 48 h at 24 °C and boiled for five minutes prior being incubated at 50 °C for 15 min in a water bath. The mixture was cooled down and filtered through muslin cloth and Whatman filter paper no. 1. The filtrates were labelled and preserved at 4 °C until further use.

### 4.2. Gas Chromatography–Mass Spectrometry (GC–MS) Analysis

GC–MS analysis of LER was employed to fractionate and identify the compounds present. For the purpose, a Thermo scientific capillary column was used with 70 eV ionization energy. MS transfer line and injector temperature was set at 220 °C. The temperature of the oven was programmed. The initial temperature was 40 °C, and 220 °C was the final temperature. The percent relative peak area was utilized to quantify all investigated identified components. The compound identification was performed on the basis of comparison mass spectra and relative retention time with Wiley library data of the GC–MS system.

### 4.3. Synthesis of Silver Nanoparticles (AgNPs)

An aqueous solution of 0.1 mM silver nitrate was mixed with 1% LER with continuous stirring for 5–10 min at 40 ± 3 °C until the color of the reaction mixture was reddish-brown via the standard method by [43]. Due to this intensive heating, the reddish-brown powder was obtained, which indicated the formation of AgNPs. These prepared NPs were analyzed through various characterizations, prior to their applications.

### 4.4. Characterization of AgNPs

The following instruments were used to determine the shape, size, composition, and morphology composition of NPs.

#### 4.4.1. Fourier Transform Infrared (FTIR) Spectroscopy

The type of associated functional groups of plant extract with nanoparticles was determined by FTIR spectroscopy. To create sample pellets, the LER and LER-AgNPs were thoroughly dried, mixed with KBr, and pressed using a hydraulic pellet press. The pellet was analyzed in an FTIR spectroscope with a resolution of 4 cm^−1^ and a scan range of 500–4000 cm^−1^ via the method of [1,44].

#### 4.4.2. UV-Vis Spectrophotometry

The UV-Vis spectrophotometry was performed to analyze the optical properties of the nanoparticles. The UV-Vis spectrophotometer was used to characterize the resultant colloidal silver nanoparticles in the wavelength range of 300 nm to 800 nm. The sample of the NPs was prepared according to the standard procedure [43].

#### 4.4.3. X-ray Diffraction (XRD)

The crystalline behavior of the prepared NPs was assessed with the help of an X-ray diffract meter (JEOL JDX 3532). The findings were derived from the atomic structure of powder samples and solid crystals, as well as the angles at which diffraction occurred. The crystalline size of the silver nanoparticles was determined using the Scherrer equation.
(1)$$D=\frac{k\lambda }{\beta Cos\theta }$$
where

D = average crystalline domain size perpendicular to reflecting planes,k = shape factor,λ = X-ray wavelength,β = FWHM (full width at the half maximum),and θ = the diffraction angle.

#### 4.4.4. Scanning Electron Microscopy (SEM) and Energy Dispersive X-ray (EDX)

SEM and EDX analyses were practiced to determine the scale, shape, and chemical composition of the synthesized NPs. A drop of the LER-AgNPs was coated on carbon tape and gold-coated with an auto fine coater (Spi-module sputter coater) before SEM analysis for morphological features. The composition of the NPs was assessed through EDX analysis.

#### 4.4.5. Dynamic Light Scattering Spectroscopy and Zeta Potential

The characterization of LER-AgNPs, involving the assessment of particulate size and the surface charge (Zeta potential), was carried out. The analysis was conducted using Photon Correlation Spectroscopy (PCS) with a Zeta Sizer Nano instrument (Malvern Instrument Ltd. ZS-90, Malvern, UK) at a scattering angle of 90° and a temperature of 25 °C.

### 4.5. Antioxidant Potential

#### 4.5.1. DPPH Scavenging Assay

Free radical scavenging assays of LER and LER-AgNPs were evaluated against 2, 2-diphenyl-1-picrylhydrazyl (DPPH). The reducing potential was measured from their bleaching action upon DPPH (purple color solution). Different grades (1 mg/mL, 2 mg/mL, and 5 mg/mL) of both LER and LER-AgNPs were prepared by mixing the respective mass per milliliter of dH_2_O with 0.1 mM DPPH, incubated in dark, and absorbency at 517 nm was measured at the first, second, and third hours of incubation. The untreated 0.1 mM DPPH was run as control while ascorbic acid (100 µg/mL) was taken as a standard antioxidant drug via the standard procedure, as described by [45,46,47]. The % antioxidant potential was determined:(2)$$\%\;Antioxidant\;potential\;=\frac{Absorbance\;of\;control\left(nm\right)-Absorbance\;of\;test\;(nm)}{Absorbance\;of\;control\left(nm\right)}$$

#### 4.5.2. Hydroxyl (•OH) Scavenging Assay

The effect of LER and LER-AgNPs on the discoloration rate of phenol dye in the presence of •OH was measured as free radical scavenging potency. Different concentrations (20, 40, 60, 80, and 100 μg/mL) of both LER and LER-AgNPs in PBS (50 mM), phenol red (0.1 mM), H_2_O_2_, and FeCl_2_ (0.5 μM) were mixed and stirred for 6 h, centrifuged at 10,000 rpm, and supernatants were tested for their absorption spectra at 430 nm using UV-Vis spectrophotometer with the method described by [48,49,50]. Untreated phenol red and ascorbic acid-treated dye were run as control and standard antioxidant, respectively. The percent •OH scavenging potential was determined as
(3)$$•OH\;scavenging\left(\%\right)=\frac{Absorption\;in\;control-Absorption\;in\;test}{Absorption\;in\;control}\times 100$$

#### 4.5.3. Hydrogen Peroxide (H_2_O_2_) Scavenging Assay

The ability of the LER and LER-AgNPs to scavenge hydrogen peroxide was determined via the method described by [18,51,52], usingH_2_O_2_ (40 mM) in phosphate buffer (pH 7.4). Different concentrations (20, 40, 60, 80, and 100 μg/mL) of each, LER, AgNPs, and ascorbic acid (standard), were mixed to 0.6 mL of 40 mM H_2_O_2_ and incubated in dark for 10 min. The absorbance of the reaction mixture at 230 nm was determined against a control containing the phosphate buffer with H_2_O_2_ and no added treatment. The percentage of H_2_O_2_ scavenging was calculated using the formula
(4)$$H_{2}O_{2}\;scavenging\;(\%)=\frac{Absorption\;in\;control-Absorption\;in\;test}{Absorption\;in\;control}\times 100$$

### 4.6. Antibacterial Potential

Antibacterial activity of LER and LER-AgNPs was determined by disc diffusion method via the protocol of [53,54]. Bacterial isolates of *Bacillus pumilus*, *Staphylococcus aureus*, *E. coli*, *Pseudomonas aeruginosa*, and *Streptococcus viridians* were cultured on Petri plates with solidified nutrient broth medium. Two different concentrations (1000 µg/mL, 2000 µg/mL) of both LER and LER-AgNPs suspension were loaded on sterile discs, placed on the surface of the medium, and poured into agar wells. Streptomycin (2 mg/mL) discs were used as a standard antimicrobial agent. After 24 and 72 h of incubation, the zone of inhibition was recorded in millimeters at each plate for antibacterial data.

### 4.7. Antifungal Potential

Antifungal of LER and LER-AgNPs was determined by diffusion method according to the standard procedure [27,55]. The fungal cultures of *Aspergillus parasiticus* Speare and *Aspergillus flavus* Link were inoculated to PDB. The antibacterial activity was assessed at two different concentrations (1000 µg/mL, 2000 µg/mL) for both LER and LER-AgNPs suspension. Streptomycin (2 mg/mL) discs were used as a standard. After 72 h of incubation, the zone of inhibition was recorded in millimeters at each plate for antifungal data. The growth inhibition in each Petri plate was measured by the following formula:Fungus growth inhibition % = (C − T) × 100(5)
where C = Average fungal mycelial growth in positive control and T = Average fungal mycelial growth in treated Petri dishes.

### 4.8. Hepatoprotective Activity

The hepatoprotective efficacy of LER-AgNPs was determined by analyzing according to the previously published standard protocol [8]. Animals were divided, six animals in each group receiving normal saline (G-1), PCM at a dose of 1 g/kg of BW (G-2), silymarin at a dose of 10 mg/kg BW (G-3), LER-AgNPs at 100 mg/Kg (G-4), 200 mg/kg (G-5), and 300 mg/kg (G-6) of body weight p.o. Blood and liver were collected from anesthetized animals and subjected to liver function tests using biochemical markers such as ALT, ALP, AST, TP, etc. and histopathological studies, respectively. 

### 4.9. Gas Chromatography–Mass Spectrometry (GC–MS) Analysis

Identification and separation were performed with GC–MS, utilizing a Thermo scientific capillary column. In order to detect, 70 eV ionization energy was utilized. MS transfer line and injector temperature was set at 220 °C. The temperature of the oven was programmed. The initial temperature was 40 °C and 220 °C was the final temperature. The percent relative peak area was utilized to quantify all identified components. The compounds identification was performed on the basis of comparison mass spectra and relative retention time with Wiley library data of the GC–MS system.

## 5. Conclusions

The green synthesis of AgNPs was accomplished using latex of *Euphorbia royleana* as a bio-reducing and stabilizing agent. The AgNPs bio-reduction technique is a simple one-step procedure that is both cost-effective and environmentally beneficial. Furthermore, bio-fabricated nanoparticles showed significant antibacterial, antifungal, antioxidant, and hepatoprotective activity. Engineered NPs significantly healed the intoxicated liver caused by a high dose of paracetamol due to oxidative stress. Nanotechnology can revolutionize the current disease management. A thorough understanding of the structural characteristics of NPs, such as surface morphology, functional groups, elemental composition, scale, and active loading capability, can provide a useful guide as a starting point for the rational selection of appropriate nanoparticles.

## Data Availability

Not applicable.

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
