# Peer review of "Euphorbia royleana Boiss Derived Silver Nanoparticles and Their Applications as a Nanotherapeutic Agent to Control Microbial and Oxidative Stress-Originated Diseases"

_pharmaceuticals, 2023, doi:10.3390/ph16101413_

Round 1
Reviewer 1 Report
The authors developed biogenic silver nanoparticles with promising applications in the field of health. It is possible to find that an interesting investigation was performed, however, in general, the manuscript presentation may be improved in order to highlight the study. Some comments and suggestions are listed below:
Abstract could be shorter and more concise.
Line 43 – Please, describe LER-AgNPs
In lines 163, 164, 176, 201, for example, the authors show the concentrations of nanoparticles (w/v). Considering that the nanoparticles were obtained by biogenic route, what technique was employed to detect the concentration of the nanoparticles? It may be reported in the methods.
Dynamic light scattering (DLS) and microelectrophoresis are suggested as important characterization techniques to obtain the hydrodynamic diameter, polydispersity and surface charge of the nanoparticles. These analysis may be performed and included in the manuscript.
Section 2.7 – Even if the protocol is published in another manuscript, give a brief description of the methodology employed to evaluate the hepatoprotective activity.
Section 2.8 – The purpose of the technique GC-MS may be described in the start of the text. It is not possible to understand which samples were analyzed by the technique. The results section describes that only the latex, and not the nanoparticles, were analyzed. It would be interesting to compare the latex and the nanoparticles in order to compare the compounds and investigate which of them remain from the synthesis.
Line 242 – The chemical formula before Figure 1 is not adequate in this position and image quality may be improved. Is this the scheme 1 which is cited in line 270?
Figure 1 – Letters a and b are not shown in the figure as described in the legend. The main peaks may be pointed in the figure, maybe using arrows. The quality of the figure may be improved.
Debye-Scherrer’s equation is shown but not mentioned in Material and Methods section. It is only cited in the results of FTIR analysis. The name of the equation and its purpose may be included in Material and Methods.
In general, it is necessary to include more details about the methods.
Figure 3 – What is the difference between the two figures shown together? Please, describe in the legend, including magnification, and separate the figures with letters a and b.
Figure 6 – The third column does not match the description in the legend. It is ascorbic acid instead of DPPH.
Line 296 – The student’ T-test is cited in this section in regard to this specific assay, however symbols indicating statistical significancy were not added in the previous figures and the statistical analysis was not previously described. A section with the description of statistical analysis employed in each assay may be included in Material and Methods.
Figures 7 and 8 may be put together as 7A and 7B.
Figure 9 design may be improved
Both the terms “Ascorbic acid” and “ascorbate” are mentioned in the text. Choose one term and use it in the whole text.
It is possible to observe in figure legends that different statistical p values were employed in Hydroxyl scavenging and Hydrogen peroxide scavenging assays. Why?
Figure 10 – Statistical analysis symbols may be shown.
Figure 11 design may be improved. Statistical analysis symbols may be shown.
Figure 12 – Quality may be improved.
Figure 13 – Quality may be improved.
Discussion – A broader discussion including reports of previous studies related to the results would make the manuscript richer. Authors may include more references and discuss them.
Writing may be improved.
Author Response
Comments and Suggestions for Authors
Reviewer 1.
The authors developed biogenic silver nanoparticles with promising applications in the field of health. It is possible to find that an interesting investigation was performed; however, in general, the manuscript presentation may be improved in order to highlight the study. Some comments and suggestions are listed below:
Query 1: Abstract could be shorter and more concise.
Authors’ response: The Abstract has been modified as per suggestion.
Query 2: Line 43 – Please, describe LER-AgNPs
Author response: The authors have described and highlighted the LER-AgNPs in Line 43.
Query 3: In lines 163, 164, 176, 201, for example, the authors show the concentrations of nanoparticles (w/v). Considering that the nanoparticles were obtained by biogenic route, what technique was employed to detect the concentration of the nanoparticles? It may be reported in the methods.
Authors responses: Thank you for the suggestion. Sir in this project we just focused on the effectiveness of the NPs. We did not quantify the concentration of the nanoparticles. In our future project, we must focus on this.
Query 4: Dynamic light scattering (DLS) and microelectrophoresis are suggested as important characterization techniques to obtain the hydrodynamic diameter, polydispersity and surface charge of the nanoparticles. These analysis may be performed and included in the manuscript.
Authors responses: Thank you for the suggestion. We performed all the possible characterization that was available to us in our institute.
Query 5: Section 2.7 – Even if the protocol is published in another manuscript, give a brief description of the methodology employed to evaluate the hepatoprotective activity.
Author response: Suggested change has been done.
Query 6: Section 2.8 – The purpose of the technique GC-MS may be described in the start of the text. It is not possible to understand which samples were analyzed by the technique. The results section describes that only the latex, and not the nanoparticles, were analyzed. It would be interesting to compare the latex and the nanoparticles in order to compare the compounds and investigate which of them remain from the synthesis.
Author response: Thank you sir for the suggestion. Sir, we performed GCMS for the analysis of the plant extract to determine the bioactive compounds. For the detailed description of NPs we performed the FTIR analysis we gave confirmation of the functional groups.
Query 7: Query 2Line 242 – The chemical formula before Figure 1 is not adequate in this position and image quality may be improved. Is this the scheme 1 which is cited in line 270?
Author response: Correction has been done.
Query 8: Figure 1 – Letters a and b are not shown in the figure as described in the legend. The main peaks may be pointed in the figure, maybe using arrows. The quality of the figure may be improved.
Author response: Correction has been done.
Query 9: Query 2Debye-Scherrer’s equation is shown but not mentioned in Material and Methods section. It is only cited in the results of FTIR analysis. The name of the equation and its purpose may be included in Material and Methods.
Author response: The suggested change has been incorporated.
Query 10: In general, it is necessary to include more details about the methods.
Author response: Detail has been added to the methodology.
Query 11: Figure 3 – What is the difference between the two figures shown together? Please, describe in the legend, including magnification, and separate the figures with letters a and b.
Authors responses: Corrected.
Query 12: Figure 6 – The third column does not match the description in the legend. It is ascorbic acid instead of DPPH.
Authors responses: Correction has been done.
Query 13: Line 296 – The student’ T-test is cited in this section in regard to this specific assay, however symbols indicating statistical significancy were not added in the previous figures and the statistical analysis was not previously described. A section with the description of statistical analysis employed in each assay may be included in Material and Methods.
Author response: Suggested change has been done.
Query 14: Figures 7 and 8 may be put together as 7A and 7B.
Author response: Suggested change has been made.
Query 15: Figure 9 design may be improved
Author response: The authors have improved the figure 9.
Query 16: Both the terms “Ascorbic acid” and “ascorbate” are mentioned in the text. Choose one term and use it in the whole text.
Author response: Ascorbate has been replaced with ascorbic acid in the whole manuscript.
Query 17: It is possible to observe in figure legends that different statistical p values were employed in Hydroxyl scavenging and Hydrogen peroxide scavenging assays. Why?
Author response: It hs been corrected in the revised version.
Query 18: Figure 10 – Statistical analysis symbols may be shown.
Author response: Done.
Query 19: Figure 11 designs may be improved. Statistical analysis symbols may be shown.
Author response: Suggested change has been incorporated.
Query 20: Figure 12 – Quality may be improved.
Author response: The authors have improved the quality of figure 12.
Query 21: Figure 13 – Quality may be improved.
Author response: The authors have improved the quality of figure 13.
Query 22: Discussion – A broader discussion including reports of previous studies related to the results would make the manuscript richer. Authors may include more references and discuss them.
Author response: The discussion has been modified has per the suggestion.
Query 23: Writing may be improved.
Authors’ response: One of our English co-author have improved the English and writing quality of the manuscript Now we hope it will be good.

Reviewer 2 Report
This study topic is interesting and important. Overall, this report has good quality and the authors have provided some results to support the significance of this study. Reasonable revisions are needed before acceptance.
Comments and suggestions:
1, the reason for choosing/designing this study need to be explained more
2, the illustration figure about this study need to be improved
3, the quality of SEM image is low, please improve if possible
4, more references about antioxidant systems are suggested to be cited/discussed, such as: Chemical Engineering Journal ,2023,472, 145061; Chinese Chemical Letters 33(4), 2022 Pages 1880-1884; Chinese Chemical Letters, 32(1), 2021, 234-238
5, more biosafety results are suggested if possible
the writing quality need to be improved
Author Response
Reviewer 2.
Comments and Suggestions for Authors
This study topic is interesting and important. Overall, this report has good quality and the authors have provided some results to support the significance of this study. Reasonable revisions are needed before acceptance.
Comments and suggestions:
- The reason for choosing/designing this study needs to be explained more
Response: The suggested change has been incorporated.
- the illustration figure about this study needs to be improved
Response: The authors have improved the figure illustration as per the suggestion.
- the quality of SEM image is low, please improve if possible
Response: The authors have improved the quality of the mentioned image.
- more references about antioxidant systems are suggested to be cited/discussed, such as: Chemical Engineering Journal ,2023,472, 145061; Chinese Chemical Letters 33(4), 2022 Pages 1880-1884; Chinese Chemical Letters, 32(1), 2021, 234-238
Response:The authors agree with the valuable suggestion of the respected reviewer and have added the more recent references about the antioxidant systems.
- more biosafety results are suggested if possible
Response: The suggested change has been incorporated.
- Comments on the Quality of English Language
the writing quality need to be improved.
Authors responses:
One of our English co-author have improved the English and writing quality of the manuscript.

Reviewer 3 Report
The paper is well-written and provides a lot of scientific information.
I have a few minor suggestions on the manuscript as following
1. Please briefly add the ecological implications of the species.
2. Elaborate the objectives of the study at the end of the introduction.
3. In the discussion section, please discuss the context of potential practical applications of the synthesized LER-AgNPs with recent literature.
After these suggestions are incorporated, I agree to accept the manuscript for publication.
The English is fine
Author Response
Reviewer 3.
Comments and Suggestions for Authors
The paper is well-written and provides a lot of scientific information.
I have a few minor suggestions on the manuscript as following
- Please briefly add the ecological implications of the species.
Author response:
Thank you for the suggestion. Suggested change has been made.
- Elaborate the objectives of the study at the end of the introduction.
Authors responses:
The authors respect the valuable suggestion of the reviewer and have elaborated the study objectives at the end of the introduction.
- In the discussion section, please discuss the context of potential practical applications of the synthesized LER-AgNPs with recent literature.
Authors responses:
The authors have added the new references of these years in the discussion section. The authors also discussed the potential practical applications of the synthesized LER-AgNPs with recent years’ references.
After these suggestions are incorporated, I agree to accept the manuscript for publication.
Comments on the Quality of English Language
The English is fine.
Thank you for the valuable suggestions.

Round 2
Reviewer 1 Report
The authors made some corrections in the manuscript, however there are some points to be adressed:
Query 3 – The authors declare that the concentration of the nanoparticles was not quantified, however, concentration values are shown in the text and figures, for example: Section 2.4.2 - Hydroxyl (•OH) scavenging assay – “concentrations (20, 40, 60, 80, and 100 μg/mL) of each LER and LER-AgNPs in PBS (50 mM), phenol red (0.1 mM), H2O2, and FeCl2 (0.5μM)were mixed and stirred for 6 hours, centrifuged at 10,000 rpm...” and the concentrations show in Y axis, figure 9. The concentrations were also necessary to perform the evaluation of hepatoprotective activity.
Query 6 – If GCMS analysis was performed to determine the bioactive compounds of the plant extract it would be interesting to present this analysis right in the start of the method section, after the preparation of the plant extract. In this way, the purpose of the analysis would be clearer.
Query 13 – The letters indicating statistical significancy were added in the figures, however, a description of the meaning of each letter may be included in the figure legends.
Query 14 – The suggestion was to put the figures regarding “Hydroxyl scavenging potential” together. Now the distribution of figures 7A, 7B and 8 are confusing.
Query 16 – The term “ascorbate” remains in figure 8
Query 17 - The legend of figure 8 indicates p < 0.01 while the legend of figure 9 indicates p<0.05.
Figure 6 – DLS data may be imported in order to prepare new figures instead of using those provided by the software.
Figure 11 – Scientific names of the fungi may be written according to scientific nomenclature rules
Figure 13 – Figure quality may be improved
Author Response
Query 3 – The authors declare that the concentration of the nanoparticles was not quantified, however, concentration values are shown in the text and figures, for example: Section 2.4.2 - Hydroxyl (•OH) scavenging assay – “concentrations (20, 40, 60, 80, and 100 μg/mL) of each LER and LER-AgNPs in PBS (50 mM), phenol red (0.1 mM), H2O2, and FeCl2 (0.5μM)were mixed and stirred for 6 hours, centrifuged at 10,000 rpm...” and the concentrations show in Y axis, figure 9. The concentrations were also necessary to perform the evaluation of hepatoprotective activity.
Reply: Dear sir, sorry for misperception on our part. Yes the concentrations of the nanoparticles were quantified as milligram per liter where the unit was convert to equivalent concentration of microgram per milliliter (weight/volume). As the molar mass of phenol, iron chloride and hydrogen peroxides were known so the molar concentrations (millimolar and micromolar etc.) were made by dissolving the appropriate molar mass per liter of solvents used. As for as the hepatoprotactive activity is concern, so the doses were selected as per the body weight of the experimental animals used.
Query 6 – If GCMS analysis was performed to determine the bioactive compounds of the plant extract it would be interesting to present this analysis right in the start of the method section, after the preparation of the plant extract. In this way, the purpose of the analysis would be clearer.
Reply: The methodology of GC-MS is shifted to the position as recommended.
Query 13 – The letters indicating statistical significancy were added in the figures, however, a description of the meaning of each letter may be included in the figure legends.
Reply: The description of letters used were added into the figure description.
Query 14 – The suggestion was to put the figures regarding “Hydroxyl scavenging potential” together. Now the distribution of figures 7A, 7B and 8 are confusing.
Reply: The figure 7A and 7B both are related to DPPH antioxidant assay. The figure 7A represent the temporal impact of various treatments on DPPH scavenging whereas the figure 7B represent the median effective concentration (EC50) and EC70 of various treatments used for their DPPH scavenging.
Query 16 – The term “ascorbate” remains in figure 8
Reply: Corrected
Query 17 - The legend of figure 8 indicates p < 0.01 while the legend of figure 9 indicates p<0.05.
Reply: In figure 8 the difference among the treatments were observed to be statistically highly significant (means the p value was less than 0.01) and hence multiple asterisks were used, while in case of figure 9 the difference among the treatments were statistically significant (means p values were less than 0.05 but were greater to 0.01)
Figure 6 – DLS data may be imported in order to prepare new figures instead of using those provided by the software.
Reply: Both of the figures were re-plotted
Figure 11 – Scientific names of the fungi may be written according to scientific nomenclature rules
Reply: The authorities were added to the fungal name as per scientific nomenclature.
Figure 13 – Figure quality may be improved
Reply: The figure is system generated and is in non-editable mode, still efforts were made to improve the quality of the figure.
